# Visual-linguistic Cross-domain Feature Learning with Group Attention and Gamma-correct Gated Fusion for Extracting Commonsense Knowledge

## ABSTRACT

Acquiring commonsense knowledge about entity-pairs from images is crucial across diverse applications. Distantly supervised learning has made significant advancements by automatically retrieving images containing entity pairs and summarizing commonsense knowledge from the bag of images. However, the retrieved images may not always cover all possible relations, and the informative features across the bag of images are often overlooked. To address these challenges, a Multi-modal Cross-domain Feature Learning framework is proposed to incorporate the general domain knowledge from a large vision-text foundation model, ViT-GPT2, to handle unseen relations and exploit complementary information from multiple sources. Then, a Group Attention module is designed to exploit the attentive information from other instances of the same bag to boost the informative features of individual instances. Finally, a Gamma-corrected Gated Fusion is designed to select a subset of informative instances for a comprehensive summarization of commonsense entity relations. Extensive experimental results demonstrate the superiority of the proposed method over state-of-the-art models for extracting commonsense knowledge.

## CCS CONCEPTS

• **Computing methodologies** → **Reasoning about belief and knowledge**.

## KEYWORDS

Commonsense knowledge extraction, Cross-modal learning, Large vision-language model, Cross-instance attention, Gamma-corrected gated fusion

## 1 INTRODUCTION

Understanding commonsense interactions between entities, such as causal relations, social norms, emotional responses, and spatial relations, is essential across numerous applications, including advanced search engines [19], generative dialogue systems [9], and visual question answering [11]. Large-scale commonsense knowledge bases (CKBs) such as Wikidata [33], ConceptNet [31] and

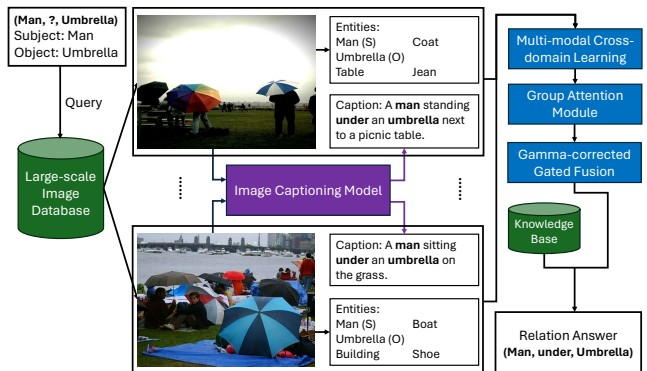

**Figure 1: The proposed model integrates linguistic features of captions generated from the large pre-trained model, ViT-GPT2, to provide the general domain knowledge on unseen relations, together with linguistic features from entity labels and visual features from retrieved images, to exploit multi-modal cross-domain interactions within individual instances and attentive support from a group of instances to extract commonsense knowledge from these sources.**

ATOMIC [27] often store abundant structured human commonsense. However, the high costs and reliance on human annotations for building these CKBs limit their scale and concept coverage [21].

Automatic Commonsense Knowledge Extraction (CKE) has received increasing research attention [13, 30, 34]. Early approaches primarily utilize text [12] or Pre-trained Language Models (PLMs) [2, 25] to obtain rich and diverse commonsense knowledge, but the inherent bias of documenting more unusual circumstances than common occurrences in textual data leads to possible over-report situations in large corpora [22, 40]. Recently, CKE from images has demonstrated promising potentials, *e.g.*, 83% of relation facts in visual relation learning datasets [34] are not covered in the large-scale text-based dataset, ConceptNet [31]. However, most image-based approaches are confined to visual commonsense in attributes such as spatial relation, color, and relative size. Recent models combine visual and textual modalities as complementary signals [34, 35], developing cross-modality learning solutions to capture more commonsense knowledge.

Despite the recent advancements, CKE still poses several challenges. 1) It often requires extracting unseen relations between entities. However, it is hard for a model to handle entities and relations not covered in the training data. Incorporating general domain knowledge could help identify unseen entity relations, while existing models often lack such general domain knowledge [8].

2) Many methods [14, 23] assume that each instance carries sufficient information to independently determine the relation of an entity pair. However, commonsense is aggregated over a group of instances, and a single instance alone is insufficient to derive a common relation. 3) The presence of inconsistent labels for the same entity relation may confuse the model to accurately derive the commonsense relations [34]. As a result, directly aggregating all the single instances that may contain noisy labels may not accurately extract common entity relations.

To better identify unseen entity relations, a Multi-modal Cross-domain Feature Learning (MCFL) framework is proposed, which integrates ViT-GPT2 [1, 29], a pre-trained large-scale visual-text model containing rich general domain knowledge, to reduce the knowledge gap between sparse objects and their rich relations. Specifically, given an entity pair, we first retrieve a bag of images containing the entities from a large image corpus, extract visual features by using VinVL [41], and derive linguistic features of entity labels through word embedding using GloVe vectors [24]. Due to the limited number of retrieved images, these features may not cover all the feasible relations between entities, while ViT-GPT2 supplements more linguistic information by generating image captions containing rich relations not seen in the retrieved images. Next, the proposed MCFL utilizes CaptionBert [40] to efficiently extract vision-text features from the pairings between visual features and entities (and captions) embeddings, as well as the linguistic feature pairings between entities and captions.

Then, to boost the features of individual instances, a Group Attention (GA) module is proposed. In single-instance learning [14, 23], each instance is treated independently to determine the relation of a specific entity pair. While effectively exploring the entity relations in an instance, this approach often overlooks the inter-instance relations and the collective insights gained from a bag of instances, which are crucial for extracting commonsense knowledge. In contrast, multi-instance learning [34] exploits the collective information from these groups to discern commonsense knowledge at the bag level, but the entity relations residing in individual instances are often overlooked. To tackle the problem, the proposed Group Attention module first derives the interaction matrix between every pair of instances, utilizes it to construct the co-attention features between every pair, and aggregates the attentive support across the whole group of instances. In such a way, the features of individual instances are greatly enhanced by exploiting partial attentive information of the whole group.

Due to automatic annotations in distant supervision with existing knowledge bases [34], instances may be assigned the wrong labels. In addition, it remains challenging to optimally aggregate common relations from multiple instances, especially when these instances are noisy. To tackle these, a Gamma-corrected Gated Fusion (GGF) module is proposed to optimally aggregate the group-attended single-instance features. Specifically, a gated net is applied to all instances to determine the adaptive weight of each instance. Intuitively, an instance with more similar instances in the group should be assigned a higher weight as it more likely contains common relations. As the gated ratios derived across many instances tend to be similar, a gamma correction is hence designed to enlarge the differences between gated ratios. As a result, the instances containing fewer occurred relations are assigned smaller

weights, while the instances containing common relations are assigned larger weights. The final relation features of all instances are fed to a multi-layer perceptron (MLP) for answer space mapping.

Our contributions can be summarized as follows. 1) To exploit unseen relations and entities from images and texts, a Multi-modal Cross-domain Feature Learning framework is proposed, which effectively integrates the general domain knowledge embedded in large pre-trained models to uncover novel relations that are missing in the retrieved images. 2) To boost the features of individual instances, a novel Group Attention module is proposed, which is capable of collaboratively exploiting the attentive support from a group of instances to enhance the description power of individual instances. 3) To adaptively select the best set of instances for extracting commonsense knowledge while mitigating noisy instances, a Gamma-Corrected Gated Fusion module is designed to assign higher weights to instances containing more common relations and alleviate the influence of noisy instances. 4) Experimental results on the large benchmark dataset demonstrate substantial performance gains of the proposed method over state-of-the-art CKE models.

## 2 RELATED WORK

Commonsense Knowledge Extraction automatically gathers implicit knowledge from diverse sources [34]. However, the construction of large-scale commonsense knowledge bases [27] heavily relies on manual annotations, greatly limiting their applications. Many models have been designed to automatically extract common relations between entities, which can be broadly divided into language-based, vision-based, and multi-modal models [34].

**Language-based CKE Models.** Early CKE models often utilize semi/unstructured text [12] or pre-trained language models [2] to identify entity relations. Schuster *et al.* [28] employed a Relation Triplet Parser (RTP) to extract relation triplets from image descriptions. Pre-trained Language Models [5] have driven the development of natural language processing, which could learn rich features from raw texts. Petroni's [25] designed a model named LAnguage Model Analysis (LAMA) to evaluate large language models such as BERT [5] for extracting commonsense knowledge. Peng *et al.* [23] applied BERT [5] on contexts and entities to predict their relations. Lin *et al.* [14] evaluated BERT [5] for encoding numerical commonsense knowledge. Lu *et al.* [18] introduced a knowledge-evolving framework by iterative consolidation and expansion with the guidance of PLMs and devised a rule generator by prompt-tuning to stimulate the rich knowledge in PLMs. Despite the progress, it has been observed that the commonsense in PLMs suffers from low consistency where small changes in queries may lead to significantly different predictions [22].

**Vision-based CKE Models.** More evidence suggests that visual perception brings commonsense that text may not reveal [34]. Yatskar *et al.* [36] uncovered general facts from image captions by leveraging WordNet and submodular k-coverage. Dai *et al.* [4] employed scene graphs to understand visual interactions between objects. But most image-based approaches are confined to visual commonsense of specific attributes [17, 40], or require large amounts of labeled data [39]. Distantly supervised learning [15, 35] has been designed to automatically create a large volume of commonsense relations without costly manual annotations. For example, Yao *et al.*

[35] employed a distant supervision strategy to align relational facts in existing knowledge bases with real-world image bags. However, distantly supervised learning often results in inaccurately labeled samples, which greatly hinders the performance of CKE models.

**Multi-modal CKE Models.** Distant supervision and utilizing information from diverse modalities are shown beneficial for commonsense extraction [7, 34]. To mitigate mislabeled instances in distant supervision, Zeng *et al.* [39] developed Piecewise Convolutional Neural Networks with multi-instance learning. Chen *et al.* [3] designed a hierarchical multi-modal fusion with a dynamic gated aggregation strategy to remove irrelevant object-text pairs. Lin *et al.* [15] utilized sentence-level attention to reduce the influence of noisy instances. Knowledge graphs are often utilized to represent entity relations. Zheng *et al.* [43] aligned different modalities through a dual graph structure, which better correlates visual relations among objects to textual relations. Feng *et al.* [7] constructed two cross-modal knowledge graphs to bridge the semantic gap and adopted a cross-attention mechanism to learn the cross-modal knowledge representations. Ma *et al.* [20] developed a multi-source knowledge reasoning graph network to learn multi-modal correlations, intra-event object relations, and inter-event semantic associations for event-centered commonsense inference. Various attention mechanisms have been designed to capture cross-modality interactions, *e.g.*, bi-linear attention [44], cross-modal attention network [42], etc. In particular, Yao *et al.* [34] utilized a pre-training vision-language model to analyze images and implemented a contrastive attention mechanism to choose descriptive images for commonsense relation summarization from entity pairs.

## 3 PROPOSED METHOD

### 3.1 Overview of Proposed Method

To tackle the challenges of commonsense knowledge extraction, Multi-modal Cross-domain Feature Learning with Group Attention and Gated Fusion (MCFL-GAGF) is proposed. The overall structure is illustrated in Fig. 2. First of all, to extract commonsense facts about a pair of entities, a bag of images containing the specified entity pair is retrieved from a large image corpus, Visual Genome [10], which contains relational triplets about entities derived from real-world images. Then, three modules are designed to extract common entity relations. 1) **Multi-modal Cross-domain Feature Learning** (MCFL) module, which extracts instance features in two stages: **Single-modal Learning** to extract visual features of the retrieved images by utilizing VinVL [41], and linguistic features of entity labels and captions generated from ViT-GPT2 [1, 29] using word embedding [40]; and **Cross-modal learning** to utilize Caption-Bert [40] to exploit interactions between modalities and modal alignment from three pairs of visual-language features. 2) **Group Attention** (GA) module, which exploits the semantic interactions across different instances to enhance the features of individual instances with the attentive support of other images in the same bag. For each instance, we first extract the co-attention features from every other instance and then aggregate these features to boost the features of the current instance. 3) **Gamma-corrected Gated Fusion** (GGF) module, which mitigates the negative effects of mislabeled relations for instances in distant supervision and optimally fuse the group-attended features of individual instances through

a set of Gamma-corrected gated ratios adaptively determined by all instance features through a set of gated nets. Finally, the discriminative commonsense knowledge derived from each instance is aggregated together with bag-level features representing collective group knowledge to produce the final prediction of entity relations.

### 3.2 Multi-modal Cross-domain Feature Learning

Given a query entity pair $(s, o)$, the commonsense relations between entities $s$ and $o$ can be summarized from a bag of $N$ images $\{I_i\}_{i=1}^N$. However, the retrieved images may not cover all the possible entity relations and it is challenging for a model to handle entities and relations that are not covered in the training data [7]. Summarizing such knowledge from diverse sources as complementary information [34] and incorporating the general domain knowledge as additional knowledge [7, 37] are two potential solutions. The proposed MCFL seamlessly integrates the two advantageous methods.

**Single-modal Learning.** Single-modal learning is designed to extract linguistic features and visual features separately. Linguistic features for entity tags are encoded by leveraging the GloVe vectors [24] as,

$$E_i = \mathcal{F}_G(\mathbb{E}_i), \tag{1}$$

where $\mathbb{E}_i$ denotes the set of $E$ entity tags for the $i$-th image and $\mathcal{F}_G$ denotes the word embedding operation in GloVe. $E_i = [e_i^1, \ldots, e_i^E]$, where $e_i^j$ is feature vector of the $j$-th entity.

Linguistic features for image captions are extracted using GloVe vectors [24], where captions are generated by ViT-GPT2 [1], a large vision-language model pre-trained with extensive image-text pairs. It contains a large amount of commonsense relations in the general domain. For each image $I_i$, a set of $W$ caption words $\mathbb{W}_i = \mathcal{F}_{VG}(I_i)$ are generated, where $\mathcal{F}_{VG}$ denotes the caption-generation operation of ViT-GPT2. $\mathbb{W}_i$ is encoded into linguistic features using GloVe vectors as,

$$C_i = \mathcal{F}_G(\mathbb{W}_i), \tag{2}$$

where $C_i = \{c_i^1, \ldots, c_i^W\}$, and $c_i^j$ denotes the $j$-th word embedding.

Visual features are extracted using VinVL [41], a visual-language model pre-trained on the Visual Genome dataset [10],

$$V_i = \mathcal{F}_V(I_i), \tag{3}$$

where $\mathcal{F}_V$ denotes the feature extraction process in VinVL. $V_i = \{v_i^1, \ldots, v_i^E\}$, where $v_i^j$ denotes the visual vector for the $j$-th entity.

**Cross-modal Learning.** Cross-modal learning is designed to exploit the interactions between visual features and linguistic features. Single-modal features are combined into three different pairings, *i.e.*, $\{v_i, e_i\}$, $\{v_i, e_i, c_i\}$, and $\{e_i, c_i\}$, corresponding to the visual-linguistic pairing between images and entities, multi-modal cross-domain feature pairing, and cross-domain linguistic feature pairing for entities and captions, respectively. A recent vision-language model that performs well in aligning features of different modalities, CaptionBert [40], is utilized to model the complex instance-level feature interactions as,

$$h_i^{VE} = \mathcal{F}_{CB}(V_i; E_i), \tag{4}$$

$$h_i^{VEC} = \mathcal{F}_{CB}(V_i; E_i; C_i), \tag{5}$$

$$h_i^{EC} = \mathcal{F}_{CB}(E_i; C_i), \tag{6}$$

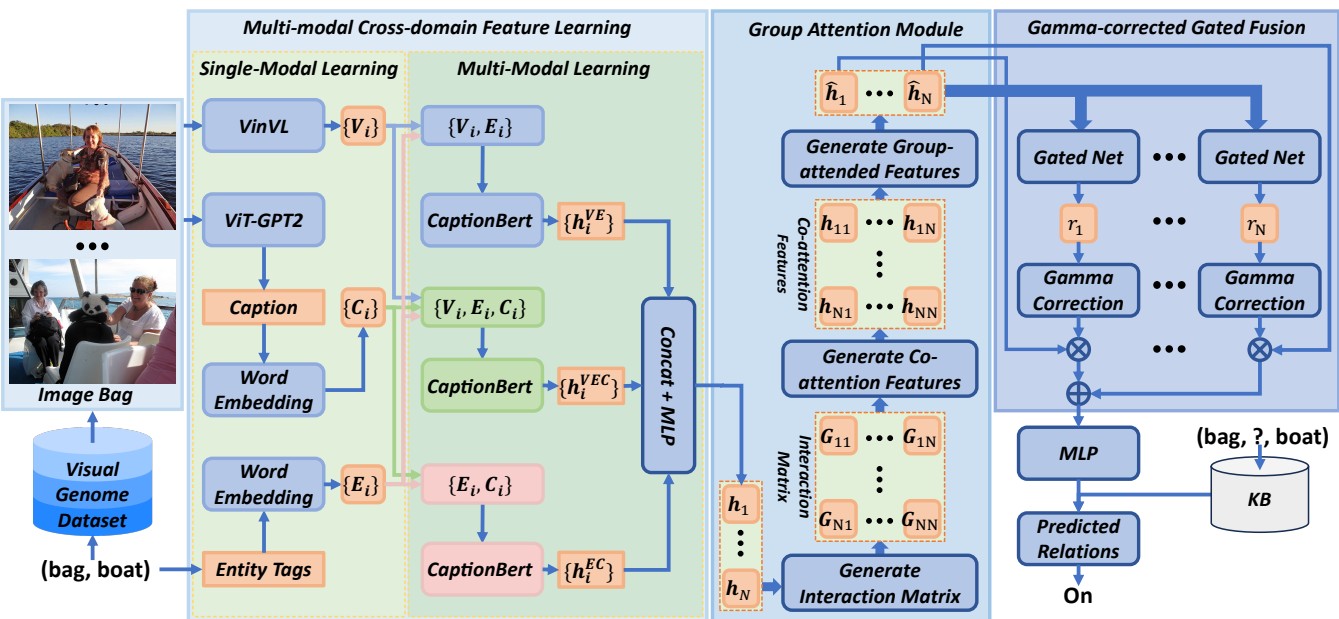

Figure 2: Overview of the proposed method, which consists of three main building blocks. 1) Multi-modal Cross-domain Feature Learning (MCFL) module to extract single-instance features by exploiting the interactions of visual and language features from multiple sources; 2) Group Attention (GA) module to boost the single-instance features by exploiting the attentive support from other instances in the same group; 3) Gamma-corrected Gated Fusion (GGF) module to optimally fuse the relational features of all instances and mitigate the noisy instances.

where $\mathcal{F}_{CB}$ represents the multi-head self-attention mechanism in CaptionBert [40]. Then, these features are jointly encoded using an MLP $\mathcal{F}_{MLP}$ as,

$$h_i = \mathcal{F}_{MLP}(h_i^{VE}; h_i^{VEC}; h_i^{EC}), \tag{7}$$

where $h_i$ are the single-instance features for the $i$-th image. The proposed MCFL incorporates the general domain knowledge from pre-trained models, extracts relation features from multiple sources, and exploits the multi-modal cross-domain interactions among these features.

### 3.3 Group Attention Module

To mitigate the mislabeled instances due to distant supervision, multi-instance learning [34] has been designed to collectively consider a group of instances and jointly exploit the information from the group to discern commonsense knowledge at the bag level. However, multi-instance learning often overlooks the discriminant information embedded in individual instances. To exploit both the discriminant information in individual instances and the collective attention information in a group of instances, a Group Attention module is proposed to extract informative features. Specifically, for each instance $h_i$, it queries every other instance $h_j$ to determine their co-attention information, where the interaction matrix $G_{ij}$ contains the attention weights indicating which part in $h_i$ is more important with respective to $h_j$,

$$G_{ij} = f_s(h_i W_G h_j^\top), \quad i, j \in \{1, ..., N\}, \tag{8}$$

where $f_s$ is the softmax function, and $W_G$ is a learnable weight matrix. The co-attention features $h_{ij}$ from $h_i$ to $h_j$ are then calculated as,

$$h_{ij} = ReLu(G_{ij}h_j), \tag{9}$$

where $ReLu$ is the ReLu function.

Finally, we derive the group-attended features as,

$$\hat{h}_i = \sigma(W_i \sum_{j=1}^{N} h_{ij} + b_i), \tag{10}$$

where $\sigma$ is the sigmoid function, $W_i$ and $b_i$ are the learnable weights and bias. The group-attended features $\hat{h}_i$ represent a dynamic focus on key common relations between entities. By exploiting the attentive information from each instance in the group, the proposed Group Attention module greatly enhances the features of individual instances.

### 3.4 Gamma-corrected Gated Fusion

Given the group-attended features of individual instances, some instances may contain more common relations while some contain less frequent relations which are potentially mislabeled. Thus, it is important to design an effective fusion scheme to uncover the common entity relations in these instances while mitigating the negative efforts of noisy instances. To achieve this, a Gamma-corrected Gated Fusion module is proposed to automatically focus on a subset of instances indicating reasonable entity relations while ignoring less informative instances. Specifically, all instance features $\{\hat{h}_i\}_{i=1}^{N}$ are fit to an MLP to excavate support evidence and

fused to construct the gated ratio $r_i$ for the $i$-th instance,

$$r_i = \sigma\left(\sum_{j=1}^{N} \boldsymbol{w}_{ij}^{\top} \hat{\boldsymbol{h}}_j + b\right), \quad i \in \{1, \ldots, N\}, \tag{11}$$

where $\boldsymbol{w}_{ij}$ and $b$ are the learnable weights and bias. It is witnessed that the derived gated ratios $\boldsymbol{r} = [r_1, \ldots, r_N]$ for different instances do not differ significantly, possibly due to the too many instances to fuse, resulting in insignificant contrast between the informative instances and less informative ones. To amplify the contrast, the Gamma correction is applied to the gated ratios. The fused features are then derived as,

$$\bar{\boldsymbol{h}} = \sum_{i=1}^{N} r_i^{\gamma} \hat{\boldsymbol{h}}_i, \tag{12}$$

where $\gamma$ is the hyper-parameter for Gamma correction. Finally, the fused features are mapped to the answer space as,

$$\bar{\boldsymbol{y}} = \sigma(\bar{W}\bar{\boldsymbol{h}}), \tag{13}$$

where $\bar{W}$ are learnable mapping weights. The proposed Gamma-corrected Gated Fusion (GGF) aims to derive an optimal subset of instances containing commonsense knowledge, which differs from the existing gated fusion [26] that aims to combine multiple feature maps into one. In addition, the proposed Gamma correction better distinguishes informative instances from non-informative ones and mitigates the latter during fusion, thereby effectively extracting commonsense knowledge. Finally, the binary cross-entropy loss is utilized in this paper,

$$\mathcal{L} = -\sum_{i=1}^{C} \bar{y}_i \log y_i + (1 - \bar{y}_i) \log(1 - y_i). \tag{14}$$

where $C$ is the number of classes, $\bar{\boldsymbol{y}} = \{\bar{y}_i\}_{i=1}^{C}$ are the predicted labels and $\boldsymbol{y} = \{y_i\}_{i=1}^{C}$ are the ground-truth labels.

## 4 EXPERIMENTAL RESULTS

### 4.1 Experimental Settings

**Dataset Description.** Very recently, Yao *et al.* [34] developed a large dataset for multi-modal commonsense knowledge extraction, which is a subset of the Visual Genome dataset [10], denoted as **VG-CKE**. The VG-CKE dataset [34] focuses on the top 100 entities and relations, comprising 6,443/1,964/678 entity pairs and 13,780/3,496/1,166 corresponding commonsense facts for training, testing, and validation, respectively. These relation facts are automatically aligned using distant supervision with image bags, resulting in 55,911 images for training, 13,722 for testing, and 5,224 for validation. Some instances may be mislabeled due to distant supervision [34].

**Compared Methods.** The proposed method is compared to nine CKE models. 1) Four language-based models. **RTP** [28] utilizes dependency trees to extract commonsense triplets from captions, whereas **LAMA** [25], **Vanilla-FT** [23] and **Prompt-FT** [14] summarize relational facts by leveraging linguistic knowledge in pre-trained language models. 2) Five multi-modal models. **ONE** [39], **ATT** [15], **AVG** [15] and **CLEVER** [34] adopt multi-instance learning to synthesize commonsense relations from multiple modalities, which extract visual features using VinVL [41] and linguistic features using Glove [24], and employ CaptionBert [40] to extract

vision-text features. They differ from the strategies of combining instance features, *e.g.*, max pooling for **ONE** [39], weighted sum for **ATT** [15], average pooling for **AVG** [15], and contrastive attention for **CLEVER** [34]. In addition, **CLEVER**$_{Ens.}$ [34] combines the results of **RTP** [28], **Vanilla-FT** [23], and **CLEVER** [34] through score-level fusion. The results of these methods are obtained from [34].

**Implementation Details.** Following [34], the bag size is set to $N = 50$. $\gamma$ in Gamma-corrected Gated Fusion is empirically set to 3. The AdamW optimizer is employed for training, with an initial learning rate of 5e-6 and a decay rate of 0.01. Training stops when no significant gain is observed in 10 consecutive epochs.

**Evaluation Metrics.** For a fair comparison, the same evaluation metrics used in [15, 34, 39] are employed, including Area Under the Curve (**AUC** and **mAUC**), F1 score (**F1** and **mF1**), as well as Precision@2% (**P@2%** and **mP@2%**), where 'm' denotes 'per-relation' macro evaluation obtained by averaging corresponding scores across different relations, and 'P@2%' represents the precision of the top 2% retrieved candidates. The AUC and F1 scores offer one-value evaluations for the model, which are principally utilized for result analysis in this study.

### 4.2 Comparison with State-of-the-Art Models

The experimental results of all the compared CKE methods on the VG-CKE dataset are summarized in Table 1. We have the following observations. 1) The proposed method significantly outperforms all compared methods in terms of all the evaluation metrics. Compared to the second-best method, CLEVER$_{Ens}$ [34], the proposed method achieves the performance gain of 3.48%, 1.37%, 1.14%, 1.71%, 1.27%, and 1.90% in terms of AUC, F1, P@2%, mAUC, mF1, and mP@2%, respectively. The large performance gains clearly demonstrate the effectiveness of the proposed method in extracting common entity relations. 2) Note that CLEVER$_{Ens}$ performs a score level fusion of three models, RTP [28], Vanilla-FT [23], and CLEVER [34]. Compared to the previous best-performing single model, CLEVER [34], the performance gains are much more significant, *e.g.*, 7.24%, 2.34%, 2.39%, 2.52%, 0.78%, and 2.68% in terms of six evaluation metrics. 3) The previous two best-performing language-based methods, Vanilla-FT [23] and Prompt-FT [14], which utilize the inherent linguistic understanding in pre-trained language models, yield a poor performance compared to the previous best performing multi-modal model, CLEVER [34]. Such observation not only demonstrates the effectiveness of integrating visual and linguistic information for CKE but also validates the design of the proposed Multi-modal Cross-domain Feature Learning framework.

The proposed method is visually compared to the previous best-performing CLEVER [34] in two scenarios. 1) The proposed model successfully uncovers the entity relations that CLEVER cannot. As shown in the first column of Fig. 3, the presence of 'man', 'under', and 'umbrella' in both captions enables the proposed method to successfully predict the 'under' relation, while CLEVER [34] fails to do so. Similar examples can be observed in the next two columns, which demonstrate the benefits of exploiting general domain knowledge in generated captions. 2) The proposed method correctly predicts the relations that CLEVER [34] inaccurately extracts. As depicted in the fourth column, the proposed method correctly

**Table 1: Comparison with state-of-the-art CKE models on the VG-CKE dataset. The proposed method significantly and consistently outperforms all the compared methods.**

| Method | AUC | F1 | P@2% | mAUC | mF1 | mP@2% |
|---|---|---|---|---|---|---|
| **RTP** (VL Workshop, 2015, [28]) | 12.30 | 23.67 | 16.65 | 4.10 | 8.62 | 7.34 |
| **LAMA** (EMNLP, 2019, [25]) | 5.97 | 14.11 | 12.80 | 3.84 | 3.59 | 5.59 |
| **Vanilla-FT** (EMNLP, 2020, [23]) | 37.28 | 47.06 | 44.21 | 17.75 | 30.98 | 17.34 |
| **Prompt-FT** (EMNLP, 2020, [14]) | 37.99 | 44.43 | 41.69 | 20.15 | 35.37 | 19.81 |
| **ONE** (EMNLP, 2015, [39]) | 19.69 | 31.10 | 25.20 | 15.70 | 30.40 | 12.82 |
| **ATT** (ACL, 2016, [15]) | 17.13 | 28.37 | 25.07 | 2.91 | 6.09 | 2.20 |
| **AVG** (ACL, 2016, [15]) | 39.04 | 47.49 | 44.34 | 24.73 | 41.07 | 20.83 |
| **CLEVER** (AAAI, 2023, [34]) | 41.92 | 48.96 | 45.84 | 26.57 | 43.62 | 22.02 |
| **CLEVER**$_{Ens.}$ (AAAI, 2023, [34]) | 45.68 | 49.93 | 47.09 | 27.38 | 43.13 | 22.80 |
| **Proposed Method** | **49.16** | **51.30** | **48.23** | **29.09** | **44.40** | **24.70** |

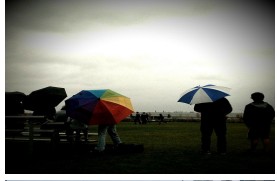
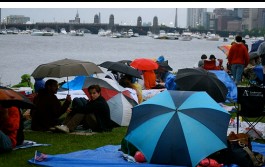
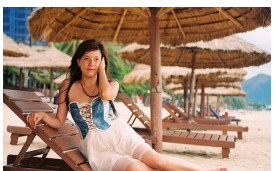
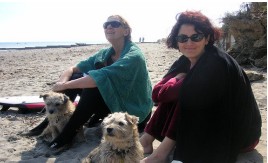
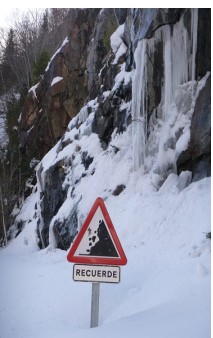
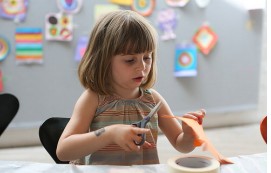
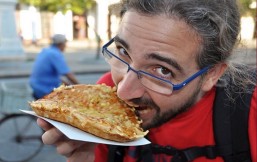
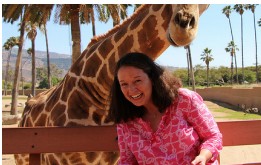
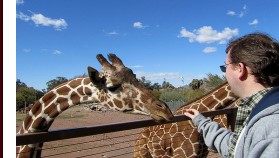

Entity Pair: (man, umbrella)
Clever: has, holding, near, with
Proposed: behind, has, holding, in, near, on, **under**, with

Caption1: A man standing under an umbrella next to a picnic table
Caption2: A man sitting under an umbrella on the grass

Entity Pair: (woman, beach)
CLEVER: in, near, on, standing on
Proposed: in, near, on, **sitting on**, standing on

Caption1: A woman sitting on a bench in front of a bench
Caption2: A woman and a dog sitting on the beach

Entity Pair: (sign, snow)
CLEVER: has, in, on, with
Proposed: above, has, in, **near**, on

Caption: A sign on a rock near a snowy mountain

Entity Pair: (paper, mouth)
CLEVER: **in**
Proposed: no relation

Caption1: A young girl cutting a piece of paper with scissors
Caption2: A person holding a slice of pizza in their hand

Entity Pair: (arm, fence)
CLEVER: near, on, **in**, **sitting on**
Proposed: near, on

Caption1: A woman standing next to a fence with a giraffe
Caption2: A giraffe is looking at the camera with its tongue out.

**Figure 3: Visual comparison with CLEVER on the VG-CKE dataset [34]. The proposed method accurately extracts more relational facts of the specified entity pairs. Facts that were missed by CLEVER but correctly summarized by the proposed methods are highlighted in green, while relation facts that are inaccurately abstracted by CLEVER are highlighted in red.**

identifies that there is no relevant relation between the 'paper' and 'mouth' entities, while CLEVER [34] wrongly identifies the relation as 'in'. Similar examples can be observed in the fifth column. The improved accuracy can be attributed to the proposed Group Attention module, which significantly enhances single-instance features, coupled with the Gamma-corrected Gated Fusion to mitigate noisy instances.

### 4.3 Ablation Studies

A set of ablation studies have been carried out to validate the effectiveness of every novel contribution. **Ablation Study of Major Modules.** An ablation study is carried out on the VG-CKE dataset [34] to evaluate the three major proposed modules. The baseline method utilizes CaptionBert [40] to extract the vision-text features from images and entities, concatenation of all instance features, and an MLP for prediction. We gradually replace the feature extraction module with the proposed Multi-modal Cross-domain

Feature Learning (MCFL), incorporate the Group Attention module (GA), and replace the concatenation with the proposed Gamma-corrected Gated Fusion (GGF). The results are summarized in Table 2. The following can be observed. 1) Compared to the baseline, by integrating the general domain knowledge embedded in captions and exploiting the interactions with other features, significant performance gains of 2.17%, 0.49%, 0.91%, and 1.02% in AUC, F1, mAUC, and mF1 are achieved, respectively. 2) By exploiting the Group Attention, the attention information across images is aggregated based on their semantic interactions and relational similarity. The AUC, F1, mAUC, and mF1 are hence further improved by 4.86%, 1.57%, 3.29%, and 6.26%, respectively, which are mainly attributed to the attentive support from other instances to enhance the instance features. 3) By incorporating the proposed Gamma-corrected Gated Fusion, the AUC, F1, mAUC, and mF1 are further boosted by 0.43%, 0.95%, 2.23% and 0.59%, respectively, as the proposed GGF better

fuses these instance features for commonsense knowledge extraction and mitigates the influence of the mislabeled instances. These ablation results demonstrate the effectiveness of all three proposed major modules.

**Table 2: Ablation study of major components of the proposed method on the VG-CKE dataset [34].**

| MCFL | GA | GGF | AUC | F1 | mAUC | mF1 |
|------|----|-----|-----|-----|------|-----|
| ✗ | ✗ | ✗ | 41.66 | 48.29 | 22.66 | 36.53 |
| ✓ | ✗ | ✗ | 43.83 | 48.78 | 23.57 | 37.55 |
| ✓ | ✓ | ✗ | 48.69 | 50.35 | 26.86 | 43.81 |
| ✓ | ✓ | ✓ | **49.16** | **51.30** | **29.09** | **44.40** |

**Ablation Study of the MCFL Module.** In the previous ablation study, it has been shown that the proposed Multi-modal Cross-domain Feature Learning module could bring significant performance gain over the baseline method. The proposed MCFL incorporates three branches of features in multi-modal learning. An ablation study is hence conducted to evaluate the effectiveness of utilizing the three branches of features. Visual features $V$ with entity embeddings $E$ serve as the baseline (Denoted as 'VE'). The combination of entity embeddings $E$ and caption embedding $C$ is denoted as 'EC' and the combination of $V$, $E$ and $C$ is denoted as 'VEC'. As shown in Table 3, the proposed MCFL brings performance gain by utilizing all three branches of multi-modal features.

**Table 3: Ablation study of the proposed MCFL module on the VG-CKE dataset [34].**

| VE | EC | VEC | AUC | F1 | mAUC | mF1 |
|----|----|-----|-----|-----|------|-----|
| ✓ | ✗ | ✗ | 41.66 | 48.29 | 22.66 | 36.53 |
| ✓ | ✓ | ✗ | 41.98 | 48.45 | 23.06 | 37.03 |
| ✓ | ✗ | ✓ | 43.55 | 48.48 | 23.25 | 37.14 |
| ✓ | ✓ | ✓ | **43.83** | **48.78** | **23.57** | **37.55** |

**Ablation Study of the Group Attention module.** The effectiveness of the Group Attention module has been verified in the ablation study in Table 2, where we demonstrate that the proposed GA effectively exploits the attentive information from a group of instances to enhance the description ability of individual instances. To further demonstrate the effectiveness of the Group Attention module, we compare it with different cross-instance methods and summarize the experimental results in Table 4. 'MLP-Instance' [38] applies MLPs on features of each individual instance to boost its description power. 'MLP-Token' [6] applies MLPs to each individual feature across instances. 'MLP-All' [32] combines the two aforementioned methods. 'BasicATT' [16] aggregates information from other instances with weights determined by feature similarity through an attention mechanism. As shown in Table 4, the proposed GA significantly surpasses all the compared methods.
**Ablation Study of Fusion Methods.** The proposed fusion method is compared with four feature-level fusion methods and two score-level fusion methods, *e.g.*, 'Feature-Concat' represents the concatenation of all instance features, and 'Feature-GatedFusion' represents gated fusion [26]. The remaining components and settings remain

**Table 4: Ablation study of Group Attention module on the VG-CKE dataset [34].**

| Cross-instance Method | AUC | F1 | mAUC | mF1 |
|-----------------------|-----|-----|------|-----|
| MLP-Instance [38] | 46.26 | 49.21 | 24.27 | 38.10 |
| BasicATT [16] | 46.33 | 49.32 | 25.00 | 38.60 |
| MLP-Token [6] | 48.01 | 49.96 | 24.06 | 36.35 |
| MLP-All [32] | 48.29 | 50.69 | 27.34 | 41.29 |
| Proposed GA | **49.16** | **51.30** | **29.09** | **44.40** |

unchanged. The ablation results are summarized in Table 5. The proposed method consistently and significantly outperforms all the compared fusion methods. The performance gains are large compared to the score-level fusion methods. In particular, compared to 'Feature-GatedFusion', it achieves performance gains of 0.93%, 0.35%, 1.13%, and 1.90% in terms of AUC, F1, mAUC, and mF1, respectively, demonstrating the effectiveness of the proposed fusion method.

**Table 5: Ablation study of various fusion methods.**

| Fusion Method | AUC | F1 | mAUC | mF1 |
|---------------|-----|-----|------|-----|
| Feature-Concat | 48.69 | 50.35 | 26.86 | 43.81 |
| Feature-GatedFusion | 48.23 | 50.95 | 27.96 | 42.50 |
| Feature-MaxPooling | 46.89 | 50.56 | 26.95 | 41.98 |
| Feature-AvgPooling | 45.86 | 50.18 | 27.21 | 42.25 |
| Score-AvgPooling | 45.79 | 49.68 | 27.66 | 43.53 |
| Score-MaxPooling | 43.24 | 48.23 | 26.81 | 42.12 |
| Proposed GGF | **49.16** | **51.30** | **29.09** | **44.40** |

## 4.4 Failure Case Analysis

We further analyze the failure cases and categorize them into two types. 1) Type-I failures comprise triplets that are not labeled in the VG-CKE dataset but can be inferred from the images. For instance, in the first case, the relational triplets (*woman*, 'in front of'/'with', *'elephant'*) are reasonably summarized by the proposed method, yet not predicted by CLEVER [34] nor included in the ground-truth annotations. In the second case, the proposed method soundly infers the relations between the 'sign' and 'fence' as 'in front of', and 'on', while CLEVER [34] only identifies the additional relationship of 'in front of'. The instances are retrieved by automatically aligning relational facts from knowledge bases to the Visual Genome dataset [10] through distant supervision, and hence there may be missing labels. Despite the challenges, the proposed method correctly identifies these relations from images. Similar cases can be observed in columns 3 to 5 in Fig. 4. 2) Type-II failures consist of incorrectly recognized relational facts by the proposed method. As shown in the first column of Fig. 5, the relation 'has', 'on', 'under', and 'with' are summarized by our model, but they are not labeled in the VG-CKE dataset. Other relational facts of 'behind' and 'of' for the entity pair (*track*, *man*), 'in' for the entity pair (*bike*, *car*), 'on' for (*person*, *book*) and 'has' for (*mountain*, *face*) are also erroneously inferred by both the proposed model and CLEVER [34], as evident from the respective images in Fig. 5.

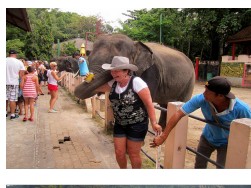
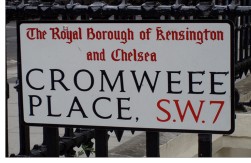
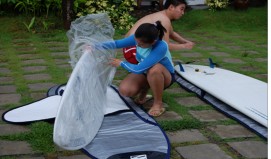
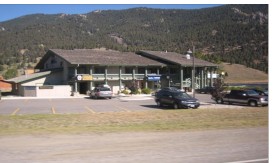
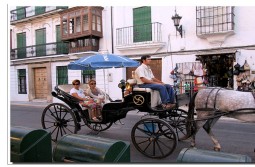

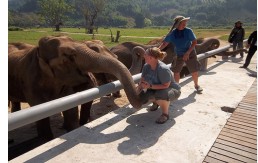
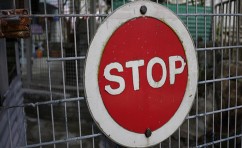
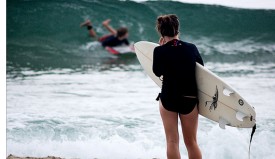
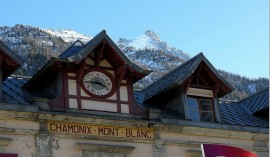
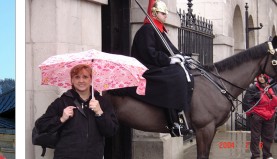

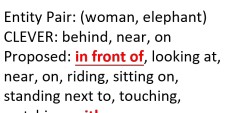
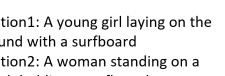

Entity Pair: (woman, elephant)
CLEVER: behind, near, on
Proposed: **in front of**, looking at, near, on, riding, sitting on, standing next to, touching, watching, **with**

Entity Pair: (sign, fence)
CLEVER: behind, has, **in front of**, near, under, with
Proposed: behind, has, **in front of**, near, **on**, under, with

Entity Pair: (surfboard, girl)
CLEVER: behind, **near**
Proposed: **attached to**, behind, **held by**, **near**, **with**

Entity Pair: (mountain, roof)
CLEVER: none
Proposed: near, **behind**

Entity Pair: (umbrella, horse)
CLEVER: behind, **near**, on
Proposed: **near**

Caption1: A man and a woman are looking at an elephant
Caption2: A woman standing next to a large elephant

Caption1: A sign on a fence that says 'no parking'
Caption2: A red and white stop sign sitting on top of a metal fence

Caption1: A young girl laying on the ground with a surfboard
Caption2: A woman standing on a beach holding a surfboard

Caption1: A small house with a bird perched on top of it.
Caption2: A large building with a clock on top of it.

Caption1: A horse pulling a carriage down a street
Caption2: A woman holding an umbrella while walking down the street

**Figure 4: Visualization of Type-I failures. The relations not labeled in the VG-CKE dataset [34] but reasonably identified by the proposed method or CLEVER are underlined. Compared to CLEVER, the proposed method discovers more new reasonable relations while minimizing the error rates.**

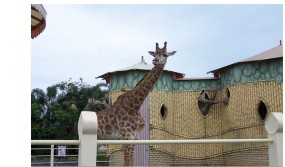
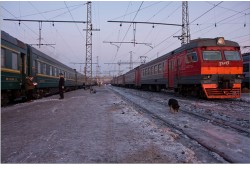
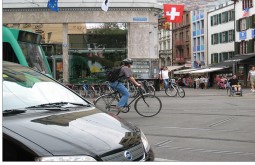
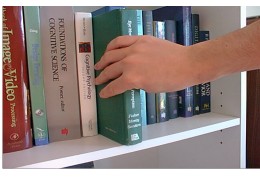
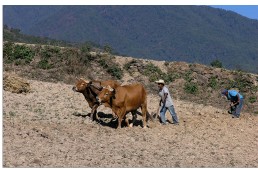

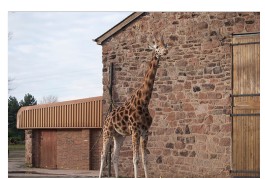
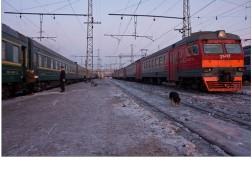
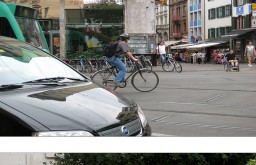
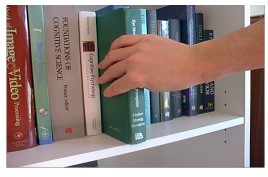
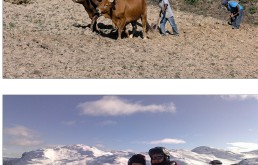

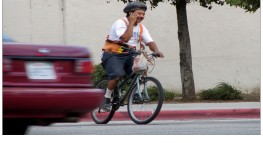
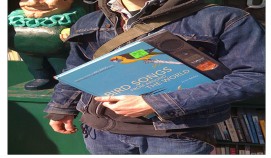

Entity Pair: (giraffe, roof)
CLEVER: near, **has**, **on**, **under**
Proposed: near, **has**, **on**, **under**, **with**

Entity Pair: (track, man)
CLEVER: **behind**, near, **of**
Proposed: **behind**, from, near, **of**

Entity Pair: (bike, car)
CLEVER: behind, **in**, in front of, near, on
Proposed: behind, **in**, in front of, near, on

Entity Pair: (person, book)
CLEVER: **has**, holding, **on**, with
Proposed: **has**, holding, **on**, with

Entity Pair: (mountain, face)
CLEVER: **has**, **with**
Proposed: **behind**, **has**

Caption1: A giraffe standing next to a fence in a zoo
Caption2: A giraffe standing next to a brick building

Caption1: A man standing next to a train on a train track
Caption2: A man riding a motorcycle on a race track

Caption1: A man riding a bike on top of a parked car
Caption2: A man riding a bike down the street next to a red car

Caption1: A person holding a book in front of a bookshelf
Caption2: A man holding a book in his hand

Caption1: A man walking a herd of cattle down a hill.
Caption2: People standing on top of a snow covered mountain

**Figure 5: Visualization of Type-II failures. The predicted relations mismatched with the ground truth are highlighted in red.**

## 5 CONCLUSION

Commonsense knowledge extraction remains challenging due to limited retrieved instances and diverse relational facts. The proposed MCFL better exploits the entity relations from multiple sources, where linguistic features extracted from captions generated by the large vision-language model ViT-GPT2 well supplement relational facts in addition to visual features of retrieved images and linguistic features of entities. These three pairs of features extract multi-modal cross-domain vision-language features. The proposed Group Attention module exploits the attentive support from other instances in the group to boost the instance features. Lastly, the proposed Gamma-correct Gated Fusion effectively aggregates all the instance features to collectively derive the commonsense relations and mitigates the mislabeled instances. Extensive experimental results on the VG-CKE dataset show the superior performance of the proposed method in commonsense knowledge extraction.

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
