# OpenReview forum: "Visual-linguistic Cross-domain Feature Learning with Group Attention and Gamma-correct Gated Fusion for Extracting Commonsense Knowledge"
_acmmm.org/ACMMM/2024/Conference — MM2024 Poster_

### Official Review · Reviewer_1c8S · 2024-05-06

**Rating:** 4
**Confidence:** 2

**Summary:**

This paper proposes a Multi-modal Cross-domain Feature Learning framework with Group Attention and Gamma-corrected Gated Fusion for extracting commonsense knowledge from images. The framework integrates visual and linguistic features, leveraging a large vision-text foundation model, ViT-GPT2, to handle unseen relations and exploit complementary information from multiple sources. A Group Attention module enhances the features of individual instances by exploiting information from other instances in the same bag, while a Gamma-corrected Gated Fusion module optimally aggregates features to extract commonsense entity relations. Experimental results demonstrate the superiority of the proposed method over state-of-the-art models.

**Strengths:**

1. The paper introduces a novel approach that integrates visual and linguistic features for commonsense knowledge extraction, addressing the limitations of existing methods.
2. The framework leverages advanced models like ViT-GPT2 and VinVL, showcasing a sophisticated approach to feature learning and fusion.
3. The paper provides extensive experimental results demonstrating the effectiveness of the proposed method, showcasing its superiority over existing models.

**Limitations:**

1. While the paper provides an overview of the proposed method and its components, a more detailed analysis of the individual modules and their impact on performance could enhance the understanding of the approach.
2. The paper focuses on commonsense knowledge extraction from images, which limits its generalizability to other domains or tasks. Exploring the applicability of the proposed framework to broader contexts could strengthen its impact.
3. While the framework achieves superior performance, the interpretability of the extracted commonsense knowledge is not explicitly addressed. Understanding how the framework generates and summarizes commonsense relations could enhance its usability.
4. The computational cost of the proposed framework, especially in terms of memory and processing requirements, is not thoroughly analyzed. This aspect is crucial for practical deployment in resource-constrained environments.
5. The paper primarily evaluates the framework on the Visual Genome dataset, which may have inherent biases or limitations. A more comprehensive evaluation on diverse datasets could provide a better understanding of the framework's robustness and generalizability.

**Suitability:**

3

---

### Official Review · Reviewer_HBZe · 2024-05-24

**Rating:** 2
**Confidence:** 2

**Summary:**

The paper introduces a novel framework for visual-linguistic cross-domain feature learning aimed at extracting commonsense knowledge from images. The proposed method, Multi-modal Cross-domain Feature Learning with Group Attention and Gamma-corrected Gated Fusion (MCFL-GAGF), leverages large-scale pre-trained models to enrich the representation of entity relations in images. The methodology is evaluated against alternatives on several benchmarks, demonstrating competitive performance in commonsense knowledge extraction.

**Strengths:**

The paper effectively combines visual and textual features using a cross-modal learning approach that integrates insights from both domains, enhancing the model’s ability to interpret complex entity relations.

The introduction of a Group Attention module allows the model to consider the context provided by other instances, providing a more nuanced understanding of each instance.

**Limitations:**

I feel quite confused about the process of generating interaction matrix and co-attention with GPT, whose motivation needs more explanation.

The complexity of the model raises my concerns, especially with the integration of multiple advanced components like group attention and gated fusion.

Given the model's sophisticated architecture, the ablation study might be better explained with more combinations.

**Suitability:**

3

---

### Official Review · Reviewer_kJe7 · 2024-05-30

**Rating:** 2
**Confidence:** 3

**Summary:**

This paper proposes a novel framework that consists of MCFL module, Group Attention module and Gamma-corrected Gated Fusion module for Commonsense Knowledge Extraction. The authors also conduct extensive experiments to evaluate the effectiveness of the proposed framework.

**Strengths:**

1. The writing is good which makes readers easy to follow.
2. The extensive experiments are conducted to demonstrate the effectiveness of the proposed framework.
3. The ablation experiments show that each part in the proposed framework is effective.

**Limitations:**

1. The technique contribution is weak.
2. Missing comparative experiments with large language models (LLMs).
3. Some ablation experiments are not comprehensive enough.

- Details:
1. What are the main differences between "Commonsense Knowledge Extraction" (CKE) in this paper and "Relation Classification in Multimodal Knowledge Graph Completion"? Both tasks supplement the relations between entity pairs. However, CKE mainly supplements location-related relations, while MKGC can supplement more complex relations, making it more meaningful.
2. The main contribution of the proposed framework is not clear. If it is about using large-scale pre-trained models to contain rich general domain knowledge, existing work [1] has already utilized PLMs to supplement relations between entity pairs. If it is about employing gate mechanism to filter noise images, the work RSME [2] for MKGC has also used gates for processing. If it is about the Group Attention Module, what is the main difference between it and self-attention?
3. In Group Attention Module, h_i will interact with all other h_j. What are the differences between it and directly performing self-attention on h_1 to h_n, and how do their performances differ?
4. The purpose of needing three types of features (VE, VC, VCE) is unclear since VCE theoretically already contains all the feature information. It is better for Table 3 to include the performance of using only VCE. Adding the two features, VE and VC, might introduce significant time consumption while providing only limited performance improvement.
5. LLMs (e.g., GPT-4) have demonstrated excellent performance across multiple tasks. It is suggested to supplement experiments related to LLMs in Table 1 to better illustrate the effectiveness of the proposed framework.

[1] Chen X, Zhang N, Li L, et al. Hybrid transformer with multi-level fusion for multimodal knowledge graph completion[C]//Proceedings of the 45th international ACM SIGIR conference on research and development in information retrieval. 2022: 904-915.
[2] Wang M, Wang S, Yang H, et al. Is visual context really helpful for knowledge graph? A representation learning perspective[C]//Proceedings of the 29th ACM International Conference on Multimedia. 2021: 2735-2743.

**Suitability:**

2

---

### Official Review · Reviewer_wTct · 2024-06-01

**Rating:** 6
**Confidence:** 3

**Summary:**

This paper proposes a multi-modal cross-domain feature learning framework aimed at improving the identification of unseen entity relations in the task of automatic commonsense knowledge extraction. Overall, the paper is well-motivated and well-written. The experimental results are promising.

**Strengths:**

The paper is well-motivated and well-written. The experimental results are promising.

**Limitations:**

Detailed comments:
1)In equation (1), why do you use Glove for word embedding? Is the word embedding consistent with the captions?
2)What is VinVL after Equation (3)?

**Suitability:**

3

---

### Meta-Review · Area_Chair_DuPX · 2024-07-03

**Recommendation:** Accept (Poster)
**Confidence:** 4

**Metareview:**

The reviewers have come up to the following strengths and limitations of the paper

STRENGTHS
- Effectiveness of the Proposed Framework: Both extensive and ablation experiments underscore the framework's effectiveness.
- Cross-Modal Learning: Integration of visual and textual features through a cross-modal learning approach is emphasized.
- Innovative Modules and Models: Introduction of novel elements like the Group Attention module and the use of advanced models (ViT-GPT2 and VinVL) are highlighted.


LIMITATIONS
- Weak Technical Contribution and Unclear Main Contribution: The main contribution of the proposed framework is not well defined, and the technical contribution is considered weak.
- Lack of Comparative Experiments: The absence of comparative experiments with large language models is noted.
- Incomplete Ablation Studies: Ablation experiments are not comprehensive enough to fully understand the impact of individual modules.
- Complexity and Motivation: Concerns about the complexity of the model and confusion regarding certain processes and motivations.
- Unclear Purpose of Features: The necessity and performance impact of using multiple types of features are questioned.
- Limited Generalizability and Dataset Bias: Focus on a specific task (commonsense knowledge extraction from images) limits generalizability, and the primary evaluation dataset (Visual Genome) may introduce biases.
- Lack of Interpretability and Computational Cost Analysis: The interpretability of the framework's outputs and the computational cost are not adequately addressed.